# Usefulness of the Measurement of Psoas Muscle Volume for Sarcopenia Diagnosis in Patients with Liver Disease

**DOI:** 10.3390/diagnostics13071245

**Published:** 2023-03-26

**Authors:** Takushi Manabe, Chikara Ogawa, Kei Takuma, Mai Nakahara, Kyoko Oura, Tomoko Tadokoro, Koji Fujita, Joji Tani, Mitsushige Shibatoge, Asahiro Morishita, Masatoshi Kudo, Tsutomu Masaki

**Affiliations:** 1Department of Gastroenterology and Hepatology, Takamatsu Red Cross Hospital, 4-1-3 Ban-cho, Takamatsu 760-0017, Japan; 2Department of Gastroenterology and Neurology, Kagawa University Faculty of Medicine, 1750-1 Ikenobe, Miki-cho, Kita-gun, Takamatsu 761-0793, Japan; 3Department of Gastroenterology and Hepatology, Kindai University Faculty of Medicine, 377-2 Ohno-higashi Osaka-sayama, Osaka 589-8511, Japan

**Keywords:** liver diseases, muscle strength dynamometer, psoas muscles, sarcopenia, SYNAPSE 3D

## Abstract

Computed tomography (CT) is often used in the diagnosis of sarcopenia. In this study, we validated the assessment of sarcopenia by the psoas muscle volume using versatile software. The study involved a retrospective analysis of data from 190 patients with liver disease who underwent grip-strength testing and abdominal pelvic computed tomography. To assess sarcopenia, SYNAPSE 3D was used to obtain the skeletal muscle index, the psoas muscle index (PMI), and the simple method. We also used the recently proposed PMI cutoff values, for which the usefulness has been evaluated (O-PMI). The cutoff value of the psoas muscle volume index (PMVI) was determined using one of the diagnostic methods as the gold standard. All diagnostic methods showed that patients with sarcopenia had shorter survival, with O-PMI having the highest hazard ratio (HR) (HR, 6.12; 95% confidence interval [CI], 2.6–14.41; *p* < 0.001). Even when sarcopenia could not be diagnosed by O-PMI, low PMVI was associated with shorter survival (HR, 3.53; 95% CI, 1.34–9.32; *p* = 0.01). PMVI may be useful in the evaluation of sarcopenia, including the identification of poor overall survival in cases that cannot be diagnosed by O-PMI, which is considered more useful than PMI.

## 1. Introduction

Sarcopenia is characterized by age-related loss of muscle mass and was first proposed by Rosenberg [1]. Recently, attention has focused on sarcopenia secondary to factors other than aging [2], and in the field of hepatology, Montano-Loza et al. reported that the prognosis of liver cirrhosis associated with sarcopenia was poor [3]. Sarcopenia has been reported to be associated with mortality and tumor recurrence in hepatocellular carcinoma (HCC) [4]. It has also been suggested that sarcopenia is associated with liver fibrosis in chronic liver disease [5]. Since then, the importance of diagnosing sarcopenia in patients with liver disease has been widely reported [4,6,7,8,9]. Recently, the Asian Working Group for Sarcopenia (AWGS) and the European Working Group on Sarcopenia have revised their diagnostic criteria and cutoff values for sarcopenia [10,11,12,13,14,15]. The second edition of the Japanese guidelines (Japan Society of Hepatology [JSH]), similarly to those of the AWGS, now requires a male grip strength of 28 kg instead of 26 kg [16]. In addition, the JSH guidelines state that computed tomography (CT) can also be used to diagnose sarcopenia, and there are numerous reports on its usefulness [17,18,19]. The diagnosis of sarcopenia by CT is generally based on the amount of skeletal muscle mass in the lumbar spine at the third lumbar (L3) level, evaluated in terms of cross-sectional area.

The volume analyzer SYNAPSE 3D (Fujifilm, Tokyo, Japan) is an image analysis system that allows users to quickly and easily obtain high-resolution three-dimensional (3D) images of organs, muscles, bones, and blood vessels using CT and MRI images taken in the past [20]. SYNAPSE 3D is popular and widely used software in Japan. Recently, servers on which SYNAPSE 3D can quickly be used from a screen of an image list used in routine medical treatment have become common, and SYNAPSE 3D, with more than 200 electronic medical records, can be used any time in hospitals for 24 h a day, and the iliopsoas muscle can easily be measured. There are also many recent reports on the usefulness of therapy support tools for HCC and the diagnosis of sarcopenia using SYNAPSE 3D [21,22,23].Therefore, in this study sarcopenia was evaluated from three-dimensional (3D) configuration data of the psoas muscle using the versatile image analysis software SYNAPSE 3D (Fujifilm, Tokyo, Japan). Recently, Ohara et al. recommended a cutoff value for psoas muscle mass of 3.74 cm^2^/m^2^ for Japanese men and 2.29 cm^2^/m^2^ for Japanese women [24]. In this study, we validated this cutoff value by comparing it with the conventional JSH diagnostic criteria.

## 2. Materials and Methods

### 2.1. Patients

This was a retrospective study of 213 patients with liver disease who underwent grip-strength measurements and abdominal pelvic CT within 2 months before and after the date of the measurements at Takamatsu Red Cross Hospital from February 2018 to May 2022. Of the 213 patients in the study, 190 were validated after excluding those who had been under observation for less than 2 months or who did not die from liver-related causes, such as hepatocellular carcinoma (HCC) progression or liver failure. The diagnosis of cirrhosis was based on CT imaging findings, platelet count, the Fibrosis-4 index, esophagogastric varices, and the presence of collateral blood vessels. HCC was diagnosed histologically or radiologically.

### 2.2. Sarcopenia

On the basis of the JSH guidelines (2nd edition), patients who met the criteria for grip strength and skeletal muscle index (SMI), psoas muscle index (PMI), and a simple method (described later in the text) were diagnosed with sarcopenia. Grip strength was measured using a Smedley grip dynamometer in accordance with the new muscle strength test guidelines of the Japanese Ministry of Education. The cutoff values for grip strength were 28 kg for men and less than 18 kg for women [16]. SMI was calculated by dividing the total muscle mass at the L3 level (rectus abdominis, transverse abdominis, internal oblique, external oblique, psoas, quadratus lumborum, and erector spinae muscles) by the square of the height. The cutoff values for SMI were 42 cm^2^/m^2^ for men and 38 cm^2^/m^2^ for women [18]. PMI was determined by dividing the area of the iliopsoas muscle at the L3 level by the square of the height and by a simple method, which was calculated as the sum of the left and right long axis × short axis of the iliopsoas muscle at the L3 level. The cutoff values for PMI and the simple method were 6.36 cm^2^/m^2^ and 6.0 cm^2^/m^2^ for men and 3.92 cm^2^/m^2^ and 3.4 cm^2^/m^2^ for women [21,25,26]. Sarcopenia was also evaluated using diagnostic criteria in accordance with the PMI cutoff values (3.74 cm^2^/m^2^ for men and 2.29 cm^2^/m^2^ for women) proposed by Ohara et al. (O-PMI) [24].

### 2.3. Muscle Mass Measurement and Equipment

CT was performed using a 64-slice multidetector-row CT scanner (Aquilion 64, Toshiba Medical Systems, Tokyo, Japan; Discovery CT750HD, GE Healthcare, Milwaukee, WI, USA) with the following scan parameters: reconstructed slice thickness = 1 mm.

SYNAPSE 3D software version 6.4 (Fujifilm) was used to measure muscle mass. Skeletal muscle was identified and quantified within a range of −29 to 150 Hounsfield units. Using this software, values for SMI, PMI, and the simple method were calculated (Figure 1). This software can also automatically measure the volume of the psoas major muscle. In this study, we evaluated sarcopenia using the psoas muscle volume index (PMVI) as well as SMI and PMI. The volume of the psoas muscle divided by the cube of the height was used to calculate PMVI.

### 2.4. Statistical Analysis

All values are expressed as the mean ± standard deviation. Overall survival (OS) was defined as the interval between the date of grip-strength measurement and the date of death from liver disease, and data were censored at the last follow-up. Kaplan–Meier curves and the log-rank test were used to determine OS, and a Cox proportional hazards model was used to evaluate the associated factors. The method with the highest hazard ratio (HR) among SMI, PMI, the simple method, and O-PMI diagnostic methods was used as the gold standard, and the receiver operating characteristic (ROC) curve was used to obtain the PMVI cutoff value. The optimal cutoff value on the ROC curve was determined on the basis of the Youden index. Positive diagnoses were evaluated using the area under the ROC curve (AUC). Pearson’s correlation coefficient was calculated to determine the correlation between PMVI and each diagnostic method. The correlation was examined using a fixed coefficient; *p* < 0.05 was considered statistically significant. All statistical analyses were performed using EZR (Saitama Medical Center, Jichi Medical University, Saitama, Japan), which is a graphical user interface for R (The R Foundation for Statistical Computing, Vienna, Austria) [27].

## 3. Results

Of the 213 patients in this study, 23 were excluded. The 23 cases that were excluded included 14 cases with an observation period of less than 2 months. The causes of death other than liver disease were cardiac disease in 3 cases, malignancy other than HCC in 3 cases, unknown cause in 3 cases, and cerebral hemorrhage in 1 case. The median observation period was 764.5 days. Of the 190 patients, 114 (60%) were male and 76 (40%) were female. The mean age of the patients was 69 ± 13.4 years. Mean grip strength was 29.4 kg for men and 18.6 kg for women; 150 patients had Child–Pugh class A, accounting for the majority of patients (79%). The underlying disease was cirrhosis in half of the patients (*n* = 105; 55%), and 82 (43%) patients had HCC. Stage 1 was the most common stage of HCC (*n* = 29; 15%); however, Stage 4 was observed as being present in 15 patients (8%) (Table 1).

The median (range) SMI for men was 44.9 cm^2^/m^2^ (range, 20.43–72.75) and 39.77 cm^2^/m^2^ (range 23.83–61.38) for women. The mean (range) PMI for men was 5.3 cm^2^/m^2^ (range 2.18–10.63) and 3.69 cm^2^/m^2^ (range 1.55–5.84) for women. For the simple method, the mean value was 5.95 cm^2^/m^2^ (range 2.4–12.55) for men and 4.32 cm^2^/m^2^ (range 1.5–8.05) for women. The mean volume of the psoas muscle was 295.8 cm^3^ (range, 114.03–587.79) for men and 171.12 m^3^ (range, 58.02–308.37) for women. The mean PMVI was 64.59 cm^3^/m^3^ (range, 26.33–133.26) for men and 48.19 cm^3^/m^3^ (range, 16.52–81.8) for women (Table 2).

Patients diagnosed with sarcopenia by each of the diagnostic criteria (SMI, PMI, the simple method, and O-PMI) had significantly shorter survival times than those without sarcopenia. Each diagnostic method indicated a poor prognosis with sarcopenia. For SMI, 5.75 HR; 95% confidence interval (CI), 2.6–12.67; *p* < 0.05. For PMI, 4.85 HR; 95% CI, 2.04–11.54; *p* < 0.05. For the simple method, 5.63 HR; 95% CI, 2.52–12.59; *p* < 0.05. For O-PMI, 6.12 HR; 95% CI, 2.6–14.41; *p* < 0.05 (Figure 2). Furthermore, O-PMI showed a higher HR than that for the PMI cutoff value in the JSH guidelines (2nd edition).

Of the four sarcopenia diagnostic methods used in this study, O-PMI had the highest HR. Therefore, we used O-PMI as the gold standard to obtain a cutoff value for PMVI and evaluated the results using that cutoff value. The optimal cutoff value for PMVI was 55.422 cm^3^/m^3^ for men (sensitivity 0.938, specificity 0.796, AUC 0.906) and 31.879 cm^3^/m^3^ (sensitivity 0.959, specificity 1, AUC 0.977) for women (Figure 3). In this study, patients who met the definition of adequate grip strength as reported in the JSH guidelines and were below the optimal PMVI values were defined as sarcopenia patients. Regarding OS, patients with sarcopenia had a poor prognosis, and low PMVI was also a poor prognostic factor (HR 4.66; 95% CI, 2.18–9.99; *p* < 0.001; Figure 4). The correlations between PMVI and SMI, PMI, and the simple method were strong, and the correlation between PMVI and PMI was the strongest (Figure 5). We also evaluated the impact of PMVI and the combination of PMVI and O-PMI among the four diagnostic methods. First, when the patients were divided into those with and without sarcopenia on the basis of O-PMI, the HR was even higher when the sarcopenia criteria were also met for PMVI compared with O-PMI alone (HR, 7.65; 95% CI, 2.95–19.83; *p* < 0.001; Figure 6a). Additionally, in patients without sarcopenia on the basis of O-PMI, OS was shorter in patients with a low PMVI compared to those with a high PMVI (HR, 3.53; 95% CI, 1.34–9.32; *p* = 0.01; Figure 6b). However, there was no significant difference between patients diagnosed with sarcopenia by PMVI and those diagnosed with or without sarcopenia by O-PMI (Figure 6c). Furthermore, there was no significant difference between patients diagnosed with sarcopenia by O-PMI and those diagnosed with or without sarcopenia by PMVI (Figure 6d).

## 4. Discussion

Sarcopenia is a subject of interest in not only liver disease but also many other conditions [28,29,30,31]. Therefore, the diagnosis of sarcopenia is important. Recently, sarcopenia guidelines have changed the diagnostic criteria and cutoff values for sarcopenia. In the AWGS regarding grip strength, the cutoff for women remained the same, but that for men increased from 26 kg to 28 kg [10]. Additionally, facilities without medical equipment such as dual-energy X-ray absorptiometry and bioelectrical impedance analysis (BIA) can now diagnose the possibility of sarcopenia based on case extraction, grip strength, and gait speed [11]. The JSH guidelines state that BIA and CT can be used to diagnose sarcopenia [18]. However, there are some caveats when measuring muscle mass with BIA. First, BIA generally cannot be performed in patients with an implanted pacemaker (PM) or implantable cardioverter defibrillator [32], and the present study included cases with implanted PMs. There are reports stating that BIA can be performed safely in patients with implanted PMs, and the number of patients with PMs and implantable cardioverter defibrillators in daily practice is increasing; therefore, further validation is needed [33,34]. In the case of end-stage cirrhosis, which is often complicated by sarcopenia, edema and ascites are common barriers to the acquisition of accurate BIA values [18].

CT and magnetic resonance imaging are performed in patients with chronic liver disease and cirrhosis for surveillance of liver cancer; therefore, it is appropriate to evaluate sarcopenia with CT [35]. The usefulness of CT-based assessment of sarcopenia has been described in many reports [21,25,36,37,38,39]. This study reaffirmed that all of the diagnostic methods for sarcopenia using CT that are recommended by the JSH are associated with prognosis. There are also reports that SMI cannot be substituted for PMI [40]. The cutoff values for SMI and PMI in that report were 50 cm^2^/m^2^ and 5.1 cm^2^/m^2^ for men and 39 cm^2^/m^2^ and 4.3 cm^2^/m^2^ for women, which are slightly closer to the cutoff values recommended by the JSH. As reported in this study, SMI is a better prognostic marker of survival than PMI, possibly because SMI has a higher proportion of total muscle mass than PMI, and therefore SMI has a higher predictive accuracy. However, in this study, O-PMI was a better prognostic marker of survival than SMI. The cutoff value of O-PMI is approximately half that of the previous PMI cutoff value, which is probably more appropriate for Japanese. In fact, the cutoff value of PMI proposed by Ohara et al. is reported to be more useful than the cutoff value of PMI in the JSH guidelines [24]. PMI cutoff values are also required in regions other than Japan and vary from 3.2~5.9 cm^2^/m^2^ for men and from 2.6~4.0 cm^2^/m^2^ for women [41]. The JSH guidelines state that the cutoff value may be changed in the future.

In sarcopenia guidelines, the recommended assessment of muscle mass on CT is generally the cross-sectional area. The SMI, PMI, and simple method recommended by the JSH are all cross-sectional. There is also another method that uses the thickness of the psoas muscle at the level of the umbilicus, which is also a cross-sectional evaluation [42]. However, the assessment of muscle mass by cross-sectional area may be inadequate because the site of measurement varies among reports, and the vertebral level is not always the largest area [43]. In comparison, the evaluation of muscle volume with 3D images is more accurate than that with one- or two-dimensional images in assessing nodules and muscles [44,45,46,47], which is the method used in this study. Notably, there are reports of the usefulness of measurements using the volume of the psoas muscle with respect to the liver region [48]. Additionally, in areas other than liver disease, measurement of psoas muscle volume has been reported to be useful as a prognostic marker of survival and progression-free survival in patients with malignant tumors as well as in relation to postoperative complications [49,50,51,52]. There have been reports that psoas muscle volume does not play a role in the prognosis of sarcopenia, but unlike the study, psoas muscle volume was not corrected for height [53]. Correcting muscle mass for height minimizes the effects of height changes from youth to old age, but few studies have actually demonstrated the effectiveness of the correction method [54]. Recently, advances in image analysis software have made it possible to easily construct 3D images. In this study, we used the general-purpose image analysis software SYNAPSE 3D, which has been introduced in more than 1350 facilities in Japan under the name SYNAPSE VINCENT and is sold worldwide, to evaluate sarcopenia from 3D configuration data of the psoas major muscle. This software can automatically calculate the 3D volume of the psoas muscle in approximately 1–2 min and is easily calculated even by new users, a feature that makes this method highly reproducible. Furthermore, the software automatically calculates the area of the psoas muscle, making it easy to measure PMI. Additionally, because of this automatic calculation, there is no variation in values between users, and the volume calculation is also highly reproducible.

In this study we used O-PMI, which had the highest HR for OS, as the gold standard and determined the optimal cutoff value for PMVI. The results showed that as with other diagnostic methods, sarcopenia was significantly associated with shorter survival compared to the absence of sarcopenia, and sarcopenia was identified as a poor prognostic factor. SMI, PMI, and the simple method were strongly correlated with PMVI. Furthermore, the combined diagnostic criteria of PMVI and O-PMI were even more strongly related to the prognosis of sarcopenia compared with other methods. It should be noted that even in patients for whom sarcopenia was not detected by O-PMI, there was a significant difference in OS in patients with low PMVI. This finding suggests that the prognosis of sarcopenia patients not diagnosed by O-PMI could also be measured. Therefore, PMVI was considered clinically useful to detect patients who are not diagnosed with sarcopenia by conventional methods but have a poor prognosis. PMVI had a strong correlation with SMI, PMI, and the simple method and may become a new diagnostic criterion for sarcopenia.

There are several limitations in this study. Data for patients with chronic hepatitis, cirrhosis, and liver cancer were not evaluated separately in this study. However, we examined OS in the cirrhosis-only group for PMVI and found significant differences (HR, 2.86; 95% CI, 1.31–6.26; *p* = 0.008). Similarly, significant differences were observed in the HCC-only group (HR, 2.57; 95% CI, 1.07–6.2; *p* = 0.03). Although this was a single-center retrospective study, the SYNAPSE 3D software is now widely used in numerous hospitals in Japan, which may make it relatively easy to validate the results in larger studies in the future.

## 5. Conclusions

PMVI obtained with SYNAPSE 3D is simple and reproducible. PMVI may also be useful in the evaluation of sarcopenia, including the identification of poor OS in patients who cannot be diagnosed by O-PMI, which is considered more useful than PMI.

## Figures and Tables

**Figure 1 diagnostics-13-01245-f001:**
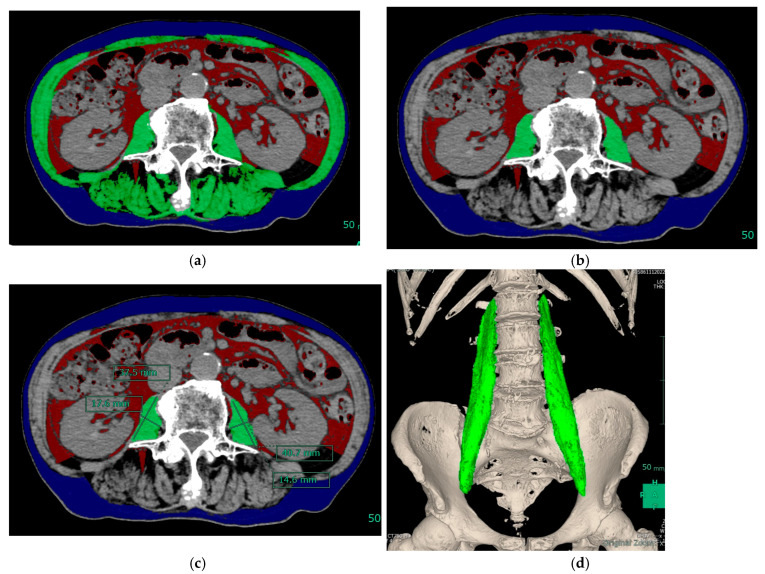
Skeletal muscle volume assessment images in a representative patient. (**a**) Abdominal CT image (horizontal section) showing the third lumbar vertebra; green indicates skeletal muscle. The skeletal muscle index was 34.34 cm^2^/m^2^. (**b**) The psoas muscle index was 3.38 cm^2^/m^2^. (**c**) The simplified method value was 4.29 cm^2^/m^2^. (**d**) The volume of the psoas major muscle was 201.89 cm^3^. Abbreviations: CT, computed tomography.

**Figure 2 diagnostics-13-01245-f002:**
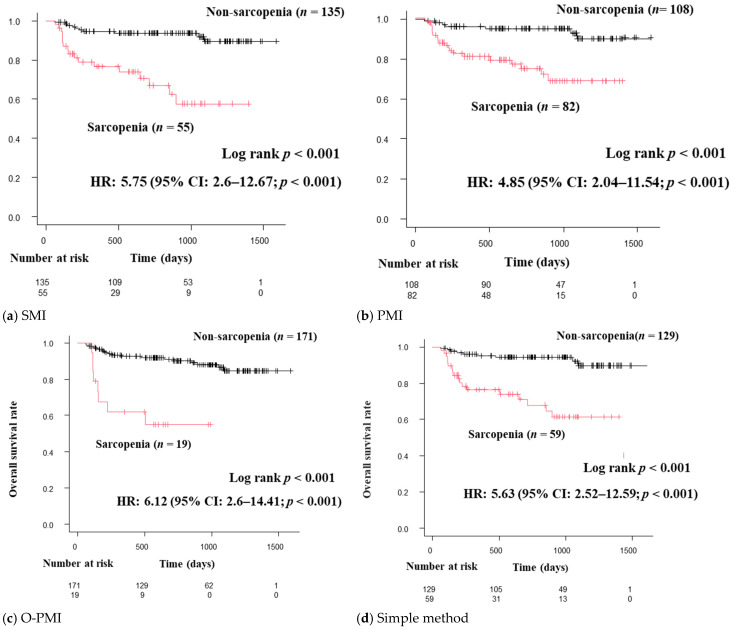
(**a**–**d**) Overall survival curves by each diagnostic method for sarcopenia. Patients with sarcopenia had significantly shorter survival compared to patients without sarcopenia for all methods. O-PMI is PMI calculated in accordance with the cutoff value proposed by Ohara et al [24]. The cutoff value was 3.74 cm^2^/m^2^ for men and 2.29 cm^2^/m^2^ for women. All diagnostic methods indicated significantly shorter survival in patients with sarcopenia compared to patients without sarcopenia. CI, confidence interval; HR, hazard ratio; PMI, psoas muscle index; SMI, skeletal muscle index.

**Figure 3 diagnostics-13-01245-f003:**
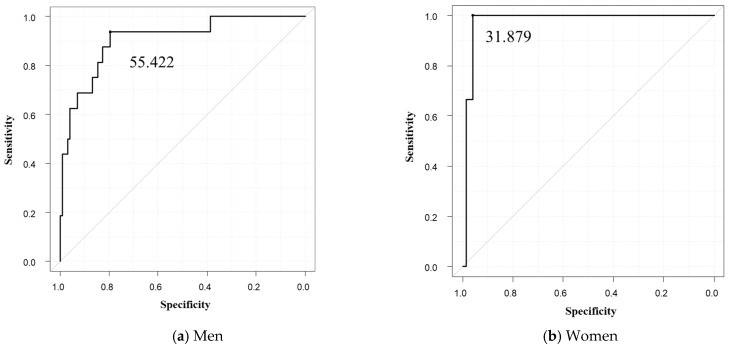
(**a**,**b**) ROC curves for PMVI by sex. The cutoff value was 55.422 cm^3^/m^3^ (sensitivity 0.938, specificity 0.796, AUC 0.906) for men and 31.879 cm^3^/m^3^ (sensitivity 0.959, specificity 1, AUC 0.977) for women. AUC, area under the ROC curve; PMVI, psoas muscle volume index; ROC, receiver operating characteristic.

**Figure 4 diagnostics-13-01245-f004:**
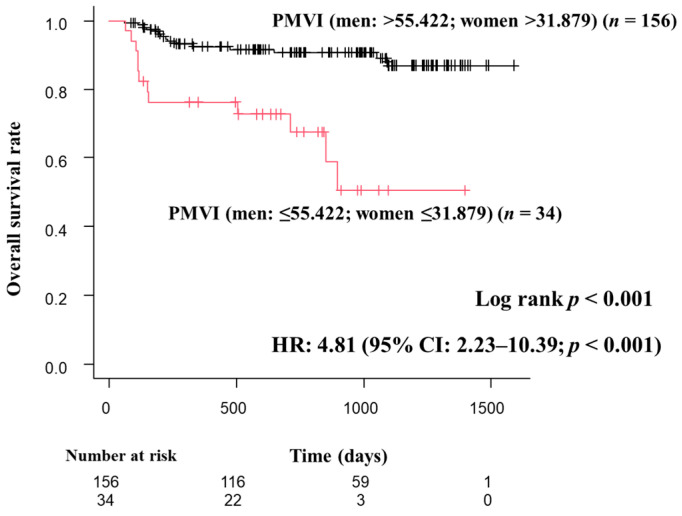
Overall survival curves by PMVI. PMVI defined sarcopenia as 55.422 cm^3^/m^3^ for men and ≤31.879 cm^3^/m^3^ for women. Patients with sarcopenia had shorter survival compared with patients without sarcopenia, and PMVI was a poor prognostic factor (HR, 4.81; 95% CI, 2.23–10.39; *p* < 0.001). CI, confidence interval; HR, hazard ratio; PMVI, psoas muscle volume index.

**Figure 5 diagnostics-13-01245-f005:**
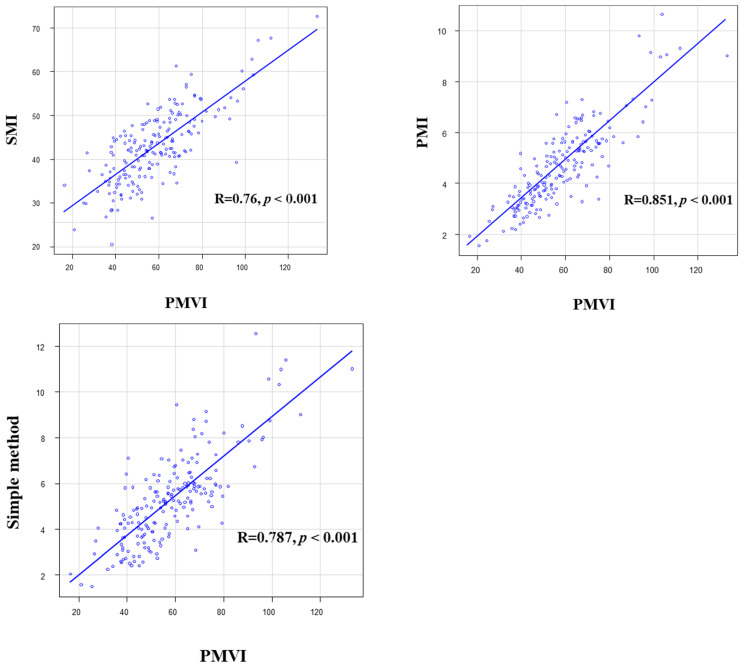
Correlations between PMVI and SMI, PMI, and the simple method. PMVI showed a strong correlation with each of the diagnostic methods. PMI, psoas muscle index; PMVI, psoas muscle volume index; SMI, skeletal muscle index.

**Figure 6 diagnostics-13-01245-f006:**
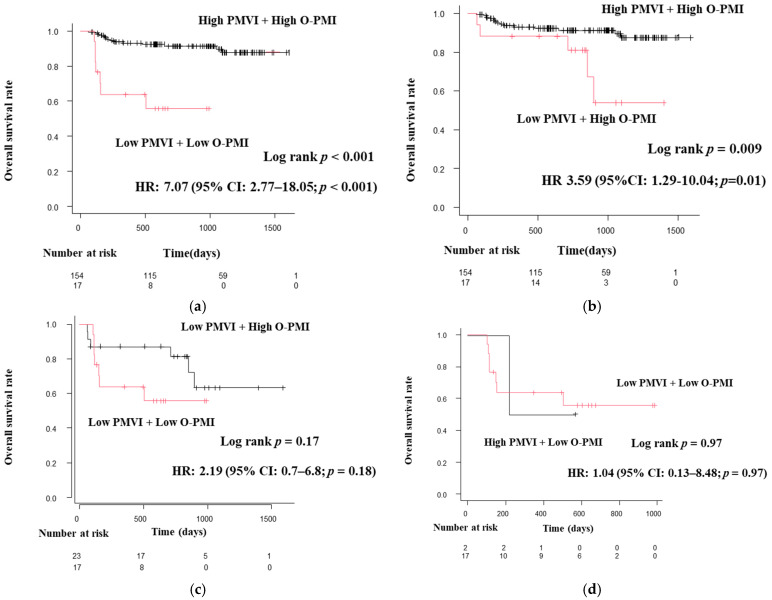
(**a**–**d**) Overall survival curves based on PMVI combined with O-PMI. Patients diagnosed with sarcopenia by O-PMI and PMVI together had a higher HR compared with patients diagnosed with sarcopenia using either method alone. Sarcopenia patients who were not diagnosed by O-PMI also had shorter survival when PMVI was low. Low O-PMI was defined as 3.74 cm^2^/m^2^ for men and ≤2.29 cm^2^/m^2^ for women; high O-PMI was defined as greater than these values. Low PMVI was defined as 54.22 cm^3^/m^3^ for men and ≤31.879 cm^3^/m^3^ for women; high PMVI was defined as greater than these values. CI, confidence interval; HR, hazard ratio; O-PMI, PMI calculated using the cutoff proposed by Ohara et al.; PMI, psoas muscle index; PMVI, psoas muscle volume index.

**Table 1 diagnostics-13-01245-t001:** Patients’ characteristics.

	*n* = 190
Men, *n* (%)	114 (60)
Mean (SD) age, years	69 (13.4)
Mean (SD) BMI, kg/m^2^	23.7 (4.3)
Mean (SD) grip strength, kg, male/female	29.4 (9.2)/18.6 (5.7)
Mean (SD) albumin, g/dL	3.8 (0.6)
Etiology: HBV/HCV/alcohol/other (%)	19/69/31/36 (10/36/16/19)
Taking diuretic: *n* (%)	31 (16)
Taking BCAA: *n* (%)	32 (17)
Taking L-carnitine: *n* (%)	6 (3)
Child–Pugh class: A/B/C: *n* (%)	150/35/5 (79/18/3)
With chronic liver disease: *n* (%)	66 (35)
With liver cirrhosis: *n* (%)	105 (55)
With hepatocellular carcinoma (HCC): *n* (%)	82 (43)
HCC stage: ½/3/4: *n* (%)	29/25/13/15 (15/13/7/8)

BCAA, branched-chain amino acid; BMI, body mass index; HBV, hepatitis B virus; HCV, hepatitis C virus; SD, standard deviation.

**Table 2 diagnostics-13-01245-t002:** Measurement results for SMI, PMI, the simple method, PMV, and PMVI.

	Men (*n* = 114)	Women (*n* = 76)
Mean (SD) SMI, cm^2^/m^2^	44.9 (8.58)	39.77 (6.9)
Mean (SD) PMI, cm^2^/m^2^	5.3 (1.59)	3.69 (0.94)
Mean (SD) simple method, cm^2^/m^2^	5.95 (1.99)	4.32 (1.4)
Mean (SD) PMV, cm^3^	295.8 (96.16)	171.12 (52.18)
Mean (SD) PMVI, cm^3^/m^3^	64.59 (1.744)	48.19 (12.47)

SMI, skeletal muscle index; PMI, psoas muscle index; PMV, psoas muscle volume; PMVI, psoas muscle volume index; SD, standard deviation.

## Data Availability

The data presented in this study are available on request from the corresponding author. The data are not publicly available due to privacy restrictions.

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
