# Peer review of "Usefulness of the Measurement of Psoas Muscle Volume for Sarcopenia Diagnosis in Patients with Liver Disease"

_diagnostics, 2023, doi:10.3390/diagnostics13071245_

Round 1

Reviewer 1 Report

Few comments:

Abstract: Line 15 and Line 191: topic--- better to change as it is not suitable e.g clinical diagnosis,...

Keywords: Line 28: Software is not suitable ward----Better to be SYNAPSE 3D.

Add an abbreviation section after the abstract.

Introduction: Line 46: SYNAPSE 3D...You should add a few sentences to explain this and how it works. Where does it use? Who can use it hepatologist or need radiologist? How much it costs? Expensive or not?

Add 1-2 sentences to explain how sarcopenia can endanger the liver patient  life.

Statistical analysis: You should calculate the sample size.

Discussion: Line 151: observed.

Line 222-23: Additionally, many reports have de- 222 scribed the usefulness of the measurement of the psoas muscle for conditions other than 223 liver disease------Like what?

Author Response

Response to reviewer 1 comments

  1. Abstract: Line 15 and Line 191: topic--- better to change as it is not suitable e.g clinical diagnosis,...

→ Thank you for your valuable comments. We have corrected the text according to your suggestion.

  1. Keywords: Line 28: Software is not suitable ward----Better to be SYNAPSE 3D.

→ Thank you for your valuable comments. We have corrected the text according to your suggestion.

  1. Add an abbreviation section after the abstract.

→ Thank you for your valuable comments. We have followed your suggestion and added it to the text.

  1. Introduction: Line 46: SYNAPSE 3D...You should add a few sentences to explain this and how it works. Where does it use? Who can use it hepatologist or need radiologist? How much it costs? Expensive or not?

→ Thank you for your valuable comments. SYNAPSE 3D can be activated from the electronic medical record by anyone, at any time. The price of SYNAPSE 3D varies greatly depending on whether it is a stand-alone version or a server type, and in the case of a server type, the price varies greatly depending on the number of contracts, the number of concurrent licenses, and the number of applications (software) to be installed. However, the fact that 1,350 facilities in Japan alone installed the system last year suggests that it is priced in line with its cost performance. We have followed your suggestion and added it to the text.

  1. Add 1-2 sentences to explain how sarcopenia can endanger the liver patient  life.

→ Thank you for your valuable comments. We have followed your suggestion and added it to the text.

  1. Statistical analysis: You should calculate the sample size.

→ Thank you for your valuable comments. In the previous study on sarcopenia and liver diseases, a cohort of 296 cases was used (Hepatol Res. 46(12), 1247-1255, 2016), and 190 cases were set as the expected number of cases that could be accumulated at our institution. Furthermore, in the ROC analysis, we considered that the number of samples required for the predicted AUC of 0.8, power of 0.9, significance level of 5%, detection rate of 90%, one-tailed test, and 2 to 1 eligible patients met the required number of samples.

  1. Discussion: Line 151: observed.

→ Thank you for your valuable comments. We have corrected the text according to your suggestion.

  1. Line 222-23: Additionally, many reports have de- 222 scribed the usefulness of the measurement of the psoas muscle for conditions other than 223 liver disease------Like what?

→ Thank you for your valuable comments. The presence of sarcopenia in malignancies other than liver is associated with survival and progression-free survival, and also with postoperative complications. We have revised the text as you indicated.

Reviewer 2 Report

It is an interesting read, predominantly region specific data. 

The following need to be added to enhance the discussion further:

a. Gu DH, Kim MY, Seo YS, et al. Clinical usefulness of psoas muscle thickness for the diagnosis of sarcopenia in patients with liver cirrhosis. Clin Mol Hepatol. 2018;24(3):319-330. doi:10.3350/cmh.2017.0077

b. Paternostro R, Lampichler K, Bardach C, Asenbaum U, Landler C, Bauer D, Mandorfer M, Schwarzer R, Trauner M, Reiberger T, Ferlitsch A. The value of different CT-based methods for diagnosing low muscle mass and predicting mortality in patients with cirrhosis. Liver Int. 2019 Dec;39(12):2374-2385. doi: 10.1111/liv.14217. Epub 2019 Sep 11. PMID: 31421002; PMCID: PMC6899596.

c. Ebadi M, Wang CW, Lai JC, Dasarathy S, Kappus MR, Dunn MA, Carey EJ, Montano-Loza AJ; From the Fitness, Life Enhancement, and Exercise in Liver Transplantation (FLEXIT) Consortium. Poor performance of psoas muscle index for identification of patients with higher waitlist mortality risk in cirrhosis. J Cachexia Sarcopenia Muscle. 2018 Dec;9(6):1053-1062. doi: 10.1002/jcsm.12349. Epub 2018 Sep 29. PMID: 30269421; PMCID: PMC6240754.

d. Bahat G, Turkmen BO, Aliyev S, Catikkas NM, Bakir B, Karan MA. Cut-off values of skeletal muscle index and psoas muscle index at L3 vertebra level by computerized tomography to assess low muscle mass. Clin Nutr. 2021 Jun;40(6):4360-4365. doi: 10.1016/j.clnu.2021.01.010. Epub 2021 Jan 16. PMID: 33516603.

The discussion needs to improved with more data to be compared .

Author Response

Response to reviewer 2 comments

  1. It is an interesting read, predominantly region specific data. 

The following need to be added to enhance the discussion further:

  1. Gu DH, Kim MY, Seo YS, et al. Clinical usefulness of psoas muscle thickness for the diagnosis of sarcopenia in patients with liver cirrhosis. Clin Mol Hepatol. 2018;24(3):319-330. doi:10.3350/cmh.2017.0077
  2. Paternostro R, Lampichler K, Bardach C, Asenbaum U, Landler C, Bauer D, Mandorfer M, Schwarzer R, Trauner M, Reiberger T, Ferlitsch A. The value of different CT-based methods for diagnosing low muscle mass and predicting mortality in patients with cirrhosis. Liver Int. 2019 Dec;39(12):2374-2385. doi: 10.1111/liv.14217. Epub 2019 Sep 11. PMID: 31421002; PMCID: PMC6899596.
  3. Ebadi M, Wang CW, Lai JC, Dasarathy S, Kappus MR, Dunn MA, Carey EJ, Montano-Loza AJ; From the Fitness, Life Enhancement, and Exercise in Liver Transplantation (FLEXIT) Consortium. Poor performance of psoas muscle index for identification of patients with higher waitlist mortality risk in cirrhosis. J Cachexia Sarcopenia Muscle. 2018 Dec;9(6):1053-1062. doi: 10.1002/jcsm.12349. Epub 2018 Sep 29. PMID: 30269421; PMCID: PMC6240754.
  4. Bahat G, Turkmen BO, Aliyev S, Catikkas NM, Bakir B, Karan MA. Cut-off values of skeletal muscle index and psoas muscle index at L3 vertebra level by computerized tomography to assess low muscle mass. Clin Nutr. 2021 Jun;40(6):4360-4365. doi: 10.1016/j.clnu.2021.01.010. Epub 2021 Jan 16. PMID: 33516603.

The discussion needs to improved with more data to be compared .

→Thank you for your valuable comments. I have referred to the cut off values of SMI and PMI in other literature and the method of measurement and compared them with this study. We agree with your opinion, and have added the corresponding part and references to the discussion section

Round 2

Reviewer 2 Report

All issues addressed

The discussion reads better now